# APD Compressible Aerogel-Like Monoliths with Potential Use in Environmental Remediation

**DOI:** 10.3390/ma12203459

**Published:** 2019-10-22

**Authors:** Hao Zhang, Fan Yang, Ruixi Bai, Zhigang Zhao, Jianguo Li, Xian Zeng, Xuesong Zhang

**Affiliations:** 1CAS Key Laboratory of Design and Assembly of Functional Nanostructures, and Fujian Provincial Key Laboratory of Nanomaterials, Fujian Institute of Research on the Structure of Matter, Chinese Academy of Sciences, Fuzhou 350002, China; xmyfzhanghao@fjirsm.ac.cn (H.Z.); bairuixi@fjirsm.ac.cn (R.B.); zgzhao@fjirsm.ac.cn (Z.Z.); lijianguo@fjirsm.ac.cn (J.L.); xmzhangxuesong@fjirsm.ac.cn (X.Z.); 2Xiamen Institute of Rare Earth Materials, Haixi Institute, Chinese Academy of Sciences, Xiamen 361021, China; 3University of Chinese Academy of Sciences, Beijing 100049, China; 4Institute of Nuclear Materials, University of Science & Technology Beijing, Beijing 100083, China

**Keywords:** aerogel, ambient pressure drying, monoliths, rare earth, absorption, environmental remediation, compressive deformation, silica, organic pollutants

## Abstract

Ambient pressure dried (APD) silica-based aerogel-like monoliths are prepared using vinyltrimethoxysilane (VTMS) as the sole silicon source by a rare-earth-assisted process. The APD method avoids the processes of solvent exchange and surface modification, is cost-effective, and reduces the preparation period from several days or weeks to 30 h. By controlling the solvent proportions, products with excellent mechanical properties, including exceptional mechanical strength and elasticity, can be synthesized. The monoliths also exhibit the outstanding characteristics of high hydrophobicity and lipophilicity and can rapidly absorb 13.5 times their weight in chloroform, showing great potential as reusable materials for application toward the separation/extraction of organic pollutants and oils.

## 1. Introduction

Aerogels, as porous materials, have great potential applications in the fields of isolation [1,2], catalysis [3,4,5], adsorption [6,7,8,9,10], and others due to their high porosity and high specific surface area. Silica aerogels are a typical representative aerogel material, and the drying process is a critical step in the preparation of aerogels for preserving their fragile pore structure because of the capillary stresses that occur at the liquid–vapor interface during the evaporation of liquid from the pores [11]. Several drying methods were developed: (1) supercritical drying (SCD), where the liquid in the pores is dislodged by a fluid in a supercritical state, which requires bypassing the critical point; (2) freeze-drying (FD), where the liquid in the pores is frozen and then sublimed, and the liquid transforms into a gas directly under vacuum, which requires bypassing the triple point; and (3) ambient pressure drying (APD), a two-step process involving solvent exchange and surface modification, which usually requires a decrease in the surface tension (*γ*) and an increase in the contact angle (*θ*), to reduce the capillary stress (*P*), according to the Laplace equation [12]:
(1)$$P=\left(-2\gamma \cdot \cos\theta \right)/r$$
where *r* is the radius of the aerogel pores. The SCD and FD methods are too costly to be widely exploited to prepare aerogels on an industrial scale; however, the APD method is suitable for large-scale industrial production due to its low cost.

Silica aerogels are synthesized by a two-step sol–gel process to obtain a wet-gel, and then the liquid is removed from the pores [13]. In a traditional preparation, the surface of the wet-gel has a number of hydroxyl groups and is highly hydrophilic; thus, the SCD method is needed to prevent structural collapse as a result of capillary stress. Recently, efforts were made to dry wet-gels to obtain aerogels by an APD method [14,15], but the processes of solvent exchange and surface modification require vast amounts of organic reagents and are time-consuming.

Rare earth (RE) are the seventeen elements in the Periodic Table, including fifteen metallic elements of the lanthanide series coupled with scandium and yttrium [16], they play critical roles in the applications of traditional and advanced materials [17], such as catalysts for petroleum refining, metal and glass additives, high-strength permanent magnets, and phosphors used in electronic displays. In this study, novel silica-based aerogel-like monoliths were fabricated by a rare-earth-assisted process via ambient pressure drying. By adding rare earth elements and controlling the amounts of ethanol and water, RE@ silica aerogel-like monoliths with excellent mechanical properties were obtained. Furthermore, the processes of solvent exchange and surface hydrophobic modification were eliminated in this synthetic route.

## 2. Materials and Methods

### 2.1. Materials

Vinyltrimethoxysilane (VTMS, C_5_H_12_O_3_Si, ≥ 98%), LaCl_3_·7H_2_O (99.9%), GdCl_3_·7H_2_O (99.9%), HoCl_3_·7H_2_O (99.9%), Rhodamine 6G (AR), and indigo carmine (96%) were purchased from Aladdin Chemistry Co., Ltd., while ethanol (AR), ammonia solution (25.0–28.0%), hydrochloric acid (AR), NaCl (AR), MgCl_2_ (AR), and FeCl_3_·6H_2_O (AR) were obtained from the Sinopharm Chemical Reagent Co., Ltd., Shanghai, China.

### 2.2. Sample Preparation and Characterization

Rare earth chlorides or other metal chlorides were dissolved in a certain proportional mixture of ultrapure water and ethanol, and then VTMS was added under continuous magnetic stirring. An HCl solution (0.1 M) was added, and vigorous stirring was maintained for another hour, to promote the hydrolysis of VTMS. Ammonia was added, the sol was transferred to a glass mold, and gelation occurred rapidly. Five hours after the wet gel had formed, it was aged in an ethanol bath at 55 °C for 12 h to strengthen its networked structure. Finally, the wet gel was directly dried in an oven at 60 °C for 8 h and 80 °C for 4 h. The synthetic process of the RE@ silica aerogel-like monoliths is shown in Scheme 1.

Bulk density was evaluated by weight and volume measurements of regular pieces of the samples. The volume of shrinkage (*V_s_*) of the product was obtained by comparing the volume of the wet-gel with the dried monoliths and was calculated by:
(2)$$V_{s}=(V_{w}-V_{a})/V_{w}\times 100\%$$
where *V_s_* was the shrinkage, *V_w_* was the volume of the wet-gel and *V_a_* was the volume of the dried aerogel.

Specific surface area was obtained by using nitrogen adsorption and the Brunauer–Emmet–Teller (BET) model (Quantachrome Autosorb iQ2, Boynton Beach, FL, USA), and the samples were degassed in a sample cell under vacuum at 110 °C for 24 h before measurement. Microstructure was observed with a scanning electron microscope (SEM, Hitachi SU1510, Tokyo, Japan), and the samples were crushed into small pieces and scattered onto sample stage with a conductive silver paste before observation. FTIR spectra (FT-IR, Nicolet is50, Thermo Fisher Scientific, Waltham, MA, USA) were obtained in the wavenumber range of 4000–425 cm^−1^, with a resolution of 4 cm^−1^ and 64 scans. The crystal structure of the monoliths was characterized by X-ray powder diffraction (XRD) (Rigaku Miniflex 600, Tokyo, Japan) with Cu Kα radiation (λ = 1.54 Å). Stress–strain curve on the uniaxial compression test was obtained by a universal material experiment machine (Instron 2365, Norwood, MA, USA), using cylindrical-shaped samples, and compression tests were conducted at a crosshead speed of 1 mm/min until the force reached 50 N. The contact angle was obtained by the sessile drop method, using high-purity water by contact angle meter (JC2000D, Beijing Zhongyi Kexin Technology Co., Ltd., China).

## 3. Results and Discussion

### 3.1. Formability Studies

VTMS was only used as a co-precursor to improve the mechanical properties of silica aerogels in previous works [18]. In this work, an ambient-pressure-dried VTMS-based single-precursor monolithic aerogel-like material was prepared by a rare-earth-assisted process. Na(I), Mg(II), and Fe(III) were selected as representative univalent, divalent, and trivalent metals, respectively, and La(III), Gd(III), and Ho(III) were selected as representative light, medium, and heavy rare earths to explore the role of rare earths in monoliths formation. All samples in Figure 1 were prepared with the same conditions, except for the type of metal ion. All of these components were prepared at least three times and showed good reproducibility. The wet-gel did not form and, so the polymer precipitates during the polycondensation process went without the addition of a metal ion; thus, a monolithic aerogel-like material cannot be obtained, and the dried product is shown to the far left in Figure 1g. Moreover, with the addition of nonrare earth metal ions, the products were cracked and fragile (Figure 1a–c). In contrast, as shown in Figure 1d–f, with the addition of rare earth ions, the products maintained a relatively intact appearance and uniform surface morphology, indicating that rare earths play a significant role in monolithic aerogel formation. SEM–EDS was employed to visualize the distribution of rare earth in the skeleton particles. The elemental distribution maps (Figure 2c) showed that rare earth was uniformly distributed. The interesting phenomenon can be explained by the fact that the hydrolyzed rare earth cations are first nucleated, and then the hydrolyzed silanes are linked around them and extend to form a three-dimensional network. The rare earth ions promote the gelation process and increase the degree of cross-linking likely due to their large ion radius, and the high coordination number increased the probability of combining with sol particles, endowing the monoliths with better formability. However, the specific mechanism of the reaction remains to be further studied in our future works. Given the rare earth reserves and preparation cost, La with large quantity and low price was selected as the additive in following experiment.

### 3.2. Physical Properties Studies

To investigate the influence of the solvent composition on the physical properties of the ambient-pressure-dried products, different proportions of water and ethanol were compared, as shown in Table 1 (all of the tests in Table 1 were conducted at three times and averaged). Samples obtained from different solvent composition were named S1 to S6 (the S is expressed as sample). It was obvious that the proportions of water and ethanol had a significant effect on the properties of the products. For S1, the solvent was only water with no ethanol, and gelation did not occur within 24 h, which could be explained by the fact that VTMS was not effectively hydrolyzed, preventing the polycondensation process. As water and VTMS are partially immiscible, additional organic solvent is needed to homogenize the mixture [11]. For S6, the solvent was only ethanol with no water, and, interestingly, phase separation did not occur in the polycondensation stage. This may be attributed to the hydrolyzation of the rare earth chlorides promoting the miscibility of VTMS in water. S6 showed minimal shrinkage and the lowest bulk density of the samples, but it appeared weak and brittle compared to S3, which also showed low shrinkage and a low density, but exhibited good flexibility, at the same time. The solvent composition for S3 was selected for the following experiments.

Brittleness is the biggest drawback for silica monoliths and greatly limits their application [19]. Traditional silica aerogels are fragile and have poor mechanical strength due to the low connectivity of its colloidal particles, which are spherical and form a pearl-necklace-like structure in which the contact area is extremely small. In this work, the particles were fused with each other to form a well-connected chain rather than the classical pearl-necklace-like structure, as proven by SEM analysis (Figure 2b), which obviously increased the contact area and endowed the products with superior mechanical strength. The uniaxial compressive stress–strain curve for VTMS-based silica aerogel-like monoliths was presented in Figure 3a, and the Young’s modulus reached 0.189 MPa. For instance, a cylindrical aerogel sample with a diameter of 3.3 cm could support a weight of 1629 g (Figure 3b) without apparent cracking. The high compressibility (Figure 3d) and flexibility (Figure 3e) of the aerogels were tested by compressing the samples and then recovering them to their original shape after the pressure was released. As shown in the FTIR spectra (Figure 2d), a number of organic functional groups existed in the original monoliths, but only the Si-O-Si bonds were preserved after thermal treatment at 800 °C. After heating, the thermally treated products lost their flexibility and were hard. The elasticity originated from the flexible organic molecular chains that repelled each other when pressed. As shown in Figure 4, the XRD patterns show a typical wide amorphous SiO_2_ peak in the range of 20°–25°, and no other characteristic peaks were observed, indicating that the monoliths are amorphous.

### 3.3. Adsorption Capacity Studies

The monoliths presented high hydrophobicity (Figure 5a) and lipophilicity (Figure 5b), and the contact angle (*θ*) reaches 138.50°, as shown in Figure 5d. The high contact angle originated in the hydrophobic groups in silica, and the thermally treated products (Figure 2d insert) lost their hydrophobic property and turned into super-hydrophilicity. Water was completely absorbed by the thermally treated products, and as soon as the water droplet falls in the test, the water droplet cannot stay on the surface of the thermally treated samples. It proves that the products turn into super-hydrophilicity after being thermally treated (Figure 5e). The hydrophobicity and excellent mechanical properties made the silica-based monoliths good absorbers for the highly efficient separation/extraction of organic pollutants and oils. The measured specific surface area is 6.41 m^2^ g^−1^ (Figure 6), which is similar to earlier reports (2.3–6.7 m^2^ g^−1^) [18]. As shown in Figure 2a, the materials exhibit the macroporous characteristic that is conducive to the APD, according to the Laplace equation (Equation (1)). The larger the pore diameter (*r*), the smaller the capillary stress (*P*) produced during solvent evaporating, so the materials are not easily fractured during the drying process. It not only plays a significant role in the preparation of materials, but also makes a great contribution to the recycling of materials, given the residual solvents in the pores after adsorption. To test the absorption capacity of the silica-based aerogel-like monoliths, two alcohols, two alkanes, two halogenated hydrocarbons, two aromatic compounds, kerosene, and acetonitrile were used. By immersing the samples in the above liquids and then immediately removing the monoliths, the absorption capacity could be quantified according to the mass of the organic liquid absorbed per unit weight of the aerogel, as shown in Figure 5c. The results indicated that the monoliths were good absorbers of organic liquid, the mass of *n*-hexane absorbed was the least of the tested solutions (5.6 times the aerogel weight), and the mass of chloroform was the greatest (13.5 times the monoliths weight). The absorption capacity of organic liquid by the samples depends on the surface tension (*γ*) of the corresponding liquid, according to Young’s equation [20]:
(3)$$\gamma_{sv}=\gamma_{sl}+\gamma_{lv}\cos\theta$$
where *γ_sv_*, *γ_sl_*, and _γlv_ are the solid–vapor, solid–liquid, and liquid–vapor interaction at the intersection of the three phases, respectively, and *θ* is the contact angle.

The products are of low-energy surfaces and the organic liquids are of low surface-tension values. Thus, the organic liquids wet the surface of samples fully when they make contact with each other, and the liquids filled up the whole empty volume of the adsorbents due to the capillary action that the silica monoliths exhibited, as they have a high absorption capacity for organic liquids.

After being absorbed, the organic liquid could be removed from the product by squeezing the sample, and the wet gel recovered its shape due to its flexibility. As shown in Figure 7a–c, all the xylenes were successfully separated from water after several cycles. The cyclic sorption–squeezing test was investigated, as shown in Figure 7d. At the first cycle, 7.18 g of ethanol was adsorbed by 1 g of aerogel-like monoliths. However, there were remnants of the ethanol in the monoliths after squeezing, due to the incomplete compression of the aerogel, which took place in order to prevent the monolithic silica materials from breaking into fragments. The silica-based aerogel-like monoliths showed stable performance during the five sorption–squeezing cycles, there was no obvious loss in adsorption capacity to allow for the fatigue resistance of the materials, and the aerogel-like material absorbed approximately seven times its dry weight in ethanol. This feature enables their prospective application as a reusable material for the rapid recovery of organic pollutants; it is very useful, especially in emergency treatment. On the other hand, it is worth noting that the possible leaching of the rare earths should be studied in our future works in order to prevent damage to the environment.

## 4. Conclusions

A novel VTMS-based aerogel-like monolith was prepared by a rare-earth-assisted process via a low-cost and rapid APD method. The products showed excellent mechanical properties, including exceptional mechanical strength and elasticity, upon adding rare earth ions with a volume ratio of EtOH/H_2_O/VTMS, which was 3:2:1. The aerogel-like monoliths also exhibited outstanding hydrophobicity and lipophilicity and could rapidly absorb 5.6–13.5 times their weight in organic liquids. The materials are an ideal reusable adsorbent for the treatment of organic pollutants due to their excellent mechanical properties and adsorption capacity.

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
