# Peer review of "APD Compressible Aerogel-Like Monoliths with Potential Use in Environmental Remediation"

_materials, 2019, doi:10.3390/ma12203459_

Round 1

Reviewer 1 Report

The paper was well articulated and written. Some minor grammar mistakes and spelling checks needs to be corrected.

The novelty is lacking though. Aerogels have been developed largely by SCD, APD and to some extend FD. The only new additions that have been generating is the use of different base precursors to form the aerogel structure. Thus, the paper does not excite me given that a lot of authors have now formulated many variants of precusors.

Nevertheless, the use of rare earth cations is a good addition to the vast list of materials used. But, it may not be commercially viable as pointed by the author given that it is not available in abundance.

A good effort overall but not a paper that blows my mind.

Author Response

Response to Reviewer 1 Comments

Point 1: The novelty is lacking though. Aerogels have been developed largely by SCD, APD and to some extend FD. The only new additions that have been generating is the use of different base precursors to form the aerogel structure. Thus, the paper does not excite me given that a lot of authors have now formulated many variants of precusors.

Response 1: Many thanks for your comment. As you said that we used of different base precursors to form the aerogel structure, we think the novelty of this paper is also shown in the following aspects:

(1) In previous reports, most aerogel structures produced by APD require a long time of solvent exchange and surface modification processes, which consume a vast number of organic solvents and take a lot of time. “The APD method avoids the processes of solvent exchange and surface modification, is cost-effective and reduces the preparation period from several days or weeks to 30 hours” (as the manuscript said in abstract, marked in yellow). Our preparation process saves a lot of time and cost, and avoids the environmental pollution caused by the large use of organic reagent.

(2) Comparing with the previous reports, our materials are very flexible, considering this special property, we try to remove the organic pollutants from the adsorbent by squeezing the sample after adsorption process, thus, the adsorbent can be reused immediately after extrusion of pollutants. It greatly improves the efficiency of product utilization, it is very useful, especially in emergency treatment.

Point 2: Nevertheless, the use of rare earth cations is a good addition to the vast list of materials used. But, it may not be commercially viable as pointed by the author given that it is not available in abundance.

Response 2: Many thanks for your comment. In this work, rare earths are used only as additives in very small amounts. Moreover, the rare earth we used was lanthanum (La), which is abundant and cheap. La is the most abundant and cheapest of all rare earths, it has long been commercialized, we compared the cost of VTMS (ï¿¥2.7265) with that of La (ï¿¥0.2770) , and found that the VTMS used in the experiment was 9.8 times more expensive than La (The materials were obtained from the Aladdin Chemistry Co., Ltd., Shang-hai, China).The amount of rare earth used is small and the price is low, in addition, considering that our materials can be reused, we think it is economical enough.

Materials

Amount

Price

Usage

Cost

LaCl3.7H2O

500 g

ï¿¥286.90

0.4828 g

ï¿¥0.2770

VTMS

500 ml

ï¿¥272.65

5 ml

ï¿¥2.7265

Reviewer 2 Report

Authors have prepared the silica-based aerogel like monoliths which were fabricated by a rare earth assisted process via ambient pressure drying. The analysis of RE@silica aerogel implemented for confirming the mechanical property, surface property and absorption property. However, this paper needs to be revised and supplemented in some places as attached below to be published by this journal.

1) Authors need to explain the synthesis of experiments by adding a process figure. Authors referred that the solvent exchange and surface modification steps were avoided. Please show the overall synthesis in more detail with including the figure. And please explain how to get a hydrophobicity was obtained without using a surface modification step?

2) Authors suggested the rare earth ions promote the gelation process and increase the degree of cross-linking and thus endow monoliths with better formability. But authors should have presented in detail about the mechanism of getting better formability by applying the rare earth elements. Also the discussion is not enough that why the authors select the La(â…¢) as rare earth element for forming the silica aerogel.

3) Authors showed the change of mechanical properties by controlling the solvent proportions. But it is hard to understand that why the polycondensation process was suppressed when the solvent was only water without ethanol. It is better to explain the mechanism in detail or cite the paper about the effect of solvents.

Reviewer 3 Report

The work entitled “Ambient Pressure Dried Compressible Aerogel-like Monoliths for Environmental Remediation” is very interesting and valuable from the practical viewpoint. In my opinion the materials were well characterized by many methods including IR and hydrophobicity measurements. Therefore the work ought to be considered for publication, however I have some comments listed below.

I suggest the authors to perform PXRD measurements. This would provide the additional structural information.

Line 128. How many times the experiment was repeated? Please provide standard deviations. The accuracy of presented results ought to be statistically analyzed.

Line 148. Please provide units for strain. I suggest to improve the quality of the plot.

Reviewer 4 Report

Reviewing of the manuscript “Ambient Pressure Dried Compressible Aerogel-like

Monoliths for Environmental Remediation”

This manuscript presents the obtaining of ambient pressure dried silica-based aerogel-like monoliths using VTMS as the sole silicon source, as well as rare earths as modifiers of the monoliths. The proposed method is faster than other procedures described for the obtaining of aerogels and also reduces the use of solvents. Moreover, the obtained monoliths show promising features of hydrophobicity and lipophylicity that can be used in the separation/extraction of organic pollutants and oils.

In general, the manuscript is well written and presents the study of promising materials for the purification of aqueous tributaries. Therefore, I do recommend its publication only after minor revisions.

Firstly, considering that the testing of the adsorption capabilities of the produced monoliths is in theoretical systems, I do recommend to modify the title of the paper in order to make it more close to reality. For instance: “APD compressible aerogel-like monoliths with potential use in environmental remediation”. This advice considers the fact that during the tests not real environmental remediation is evaluated so the scope of the paper must be presented in a more precise way.

In figure 6d, the adsorption capacity of xylene presented by  one of monoliths shows a slight decreasing. So probably more cycles are required in order to establish the stability of the adsorption capacity. If no additional experiments are considered for the present work, at least this fact has to be mentioned in the discussion and proposed for future works.

A characterization of the monoliths after the adsorption-squeezing cycles should be proposed for a better understanding of the possible modification of the structure after repeated uses.

The possible leaching of some of the components of the monoliths has to be considered and evaluated. Especially the leaching of rare earths in order to prevent damage to the environment. As in the case of the previous suggestion, this has to be at least mentioned as a subject of study in future works.

Round 2

Reviewer 2 Report

The authors have satisfactorily answered the comments and have made all the necessary changes in the revised manuscript. Therefore, I recommend publication of this revised manuscript.